# inPOSE: A Flexible Toolbox for Chromosomal Cloning and Amplification of Bacterial Transgenes

**DOI:** 10.3390/microorganisms10020236

**Published:** 2022-01-21

**Authors:** Ranti Dev Shukla, Ágnes Zvara, Ákos Avramucz, Alona Yu. Biketova, Akos Nyerges, László G. Puskás, Tamás Fehér

**Affiliations:** 1Synthetic and Systems Biology Unit, Institute of Biochemistry, Biological Research Centre of the Eötvös Lóránd Research Network, H-6726 Szeged, Hungary; dev.ranti@brc.hu (R.D.S.); avramucz.akos@brc.hu (Á.A.); alyona.biketova@gmail.com (A.Y.B.); akos_nyerges@hms.harvard.edu (A.N.); 2Doctoral School in Biology, Faculty of Science and Informatics, University of Szeged, H-6720 Szeged, Hungary; 3Laboratory for Functional Genomics, Biological Research Centre of the Eötvös Lóránd Research Network, H-6726 Szeged, Hungary; zvara.agnes@brc.hu (Á.Z.); puskas.laszlo@brc.hu (L.G.P.); 4Jodrell Laboratory, Royal Botanic Gardens, Kew, Richmond TW9 3AB, UK; 5Department of Genetics, Harvard Medical School, Boston, MA 02115, USA

**Keywords:** chromosomal gene cloning, genome editing, transgene expression, lambda-Red-mediated recombination, insertion sequence, mobile element, replicative transposition, selection-free maintenance

## Abstract

Cloning the genes and operons encoding heterologous functions in bacterial hosts is now almost exclusively carried out using plasmid vectors. This has multiple drawbacks, including the need for constant selection and variation in copy numbers. The chromosomal integration of transgenes has always offered a viable alternative; however, to date, it has been of limited use due to its tedious nature and often being limited to a single copy. We introduce here a strategy that uses bacterial insertion sequences, which are the simplest autonomous transposable elements to insert and amplify genetic cargo into a bacterial chromosome. Transgene insertion can take place either as transposition or homologous recombination, and copy number amplification is achieved using controlled copy-paste transposition. We display the successful use of IS*1* and IS*3* for this purpose in *Escherichia coli* cells using various selection markers. We demonstrate the insertion of selectable genes, an unselectable gene and a five-gene operon in up to two copies in a single step. We continue with the amplification of the inserted cassette to double-digit copy numbers within two rounds of transposase induction and selection. Finally, we analyze the stability of the cloned genetic constructs in the lack of selection and find it to be superior to all investigated plasmid-based systems. Due to the ubiquitous nature of transposable elements, we believe that with proper design, this strategy can be adapted to numerous other bacterial species.

## 1. Introduction

Today, the fastest and simplest way to maintain and/or express a gene of interest is to clone it into a bacterial plasmid, especially if one relies on the numerous technical improvements developed to increase the efficiency, speed and flexibility of this process [1]. Protein expression from plasmids, however, may significantly alter the physiology of the host cell, especially in the case of high copy numbers, thereby derailing fundamental studies [2,3]. Plasmids also require constant selection, often warranted by antibiotics and their respective resistance genes. From an industrial viewpoint, this not only poses an additional cost due to the administration of the antibiotic, but also due to the need for its removal from the final fermentation product. In addition, secreted resistance factors such as the β-lactamase enzyme allow plasmids to be lost from cells during selection, leading to inhomogeneous cell populations [4]. A similar impairment of experimental reproducibility can be caused by variations in plasmid copy numbers [5,6,7]. These factors all indicate the necessity to develop techniques permitting transgene insertion into the bacterial chromosome.

The earliest methods of modifying bacterial genomes used mobile DNA, such as Mu [8] and Tn*5* [9], to insert resistance genes along with other genes of interest into the chromosome. Subsequently, the integration of circular plasmid DNA into the bacterial genome by RecA-mediated recombination became the method of choice for genome modification [10,11]. This was eventually outcompeted by linear DNA-mediated genome engineering, often referred to as recombineering [12]. This process mostly relies on phage recombinases to carry out the double recombination step, which could be facilitated by cleaving the target DNA by homing endonucleases [13] or the CRISPR/Cas machinery [14]. A further alternative strategy was to target phage attachment sites [3,15], the Tn7 attachment site [16,17] or FRT sites [18] on the genome using the corresponding phage integrase, the Tn7 transposase or FLP recombinase enzyme, respectively, to integrate plasmid-borne transgenes.

Although multiple techniques have been developed to integrate transgenes into the bacterial genome, very few projects have succeeded in achieving this in multiple copies. The first example used the transposase of Tn*1545* to generate a library of insertions, marked with an antibiotic resistance. In a second step, the inserted cassettes were accumulated in one strain using P1 transduction, always removing the selective marker after each transfer using the *xis* and *int* functions of phage λ [19]. Another work used a phage integrase to insert the DNA cassette and the FLP/FRT system to eliminate the selection marker, thereby allowing further integrations at different phage attachment sites [20]. These two projects managed to accumulate the *lacZ* gene in the *E. coli* chromosome in three and four copies, respectively. The third strategy applied true gene copy number amplification instead of serial re-integration. The co-integrate was marked with an antibiotic resistance gene, and serially passaging cells in an increasing antibiotic concentration allowed selection for spontaneous RecA-mediated gene duplications arising from unequal crossovers between the daughter chromosomes [21]. Copy numbers up to ~40 were achieved using that strategy, which ultimately required deleting the *recA* gene of the host cell to stabilize the tandem repeats. Modifications of this strategy that apply the Cre/Lox site-specific recombinase system [22] or triclosan selection [23,24] have also been successfully used to achieve similar levels of copy number amplification.

Insertion sequences (IS elements or ISes) are the smallest autonomous mobile genetic elements found in nature [25]. They comprise only one or two genes, which are responsible for the transposition process, surrounded by inverted repeats. To date, 29 families of ISes have been described [25]. ISes belong to prokaryotic transposable elements (TEs), along with more complex members such as transposons, integrative conjugative elements, integrative mobilizable elements, type I and II introns, transposable and satellite prophages, mobile genomic islands, inteins, retrons and IStrons [26,27]. Bacterial transposons differ from ISes in the sense that in addition to transposases, they also contain genes encoding functions unrelated to transposition. Over the decades, bacterial transposons have served as popular tools of molecular biology, used for gene delivery, mutagenesis and functional genomics studies [28]. IS elements, however, are much less often used in molecular biology. Notable cases nevertheless exist: for example, IS*21* has been used for the linker-scanning mutagenesis of an enzyme-encoding gene cloned in *E. coli* [29]. Another exciting work described the reprogramming of IS*608* (IS*Hp608*) of *Helicobacter pylori* to allow predictable integration at chosen target sites, both in vitro and in vivo, years before the outbreak of the CRISPR-era [30]. Similarly, the fusion of the IS*30* transposase to various specific DNA-binding proteins permitted the directed integration of the IS*30* element both in *Salmonella* Enteritidis [31] and in zebrafish [32]. A method relying on IS*608* insertion by homologous recombination, followed by its precise transposase-mediated excision, has also been developed for editing short genomic segments [33].

In this work, we tested the application of two IS elements, IS*1* and IS*3,* for chromosomal gene cloning and amplification in *E. coli*. We reported the insertion of marked and unmarked genes into genomic ISes in one or two copies, followed by the copy number amplification of the loaded ISes, exploiting their copy-and-paste transposition.

## 2. Materials and Methods

### 2.1. Strains, Chemicals and Media

The *E. coli* strains modified in this study are listed in Table 1. Bacteria were grown in liquid Luria–Bertani medium (LB), or on LB plates containing 1.5% agar. Components of the media were obtained from Molar Chemicals Kft., Halásztelek, Hungary. Antibiotics were obtained from Sigma Aldrich (St. Louis, MO, USA) and were used in the following concentrations: chloramphenicol (Cm): 25 μg/mL; ampicillin (Ap): 100 μg/mL; kanamycin (Km): 25 μg/mL; spectinomycin (Sp): 50 μg/mL; anhydrotetracycline (aTc): 50 ng/mL. Antibiotic-gradient plates were made applying the protocol of Szybalski and Bryson [34], but using 60× higher Sp and 20× higher Km concentrations in the top layer, compared to the values listed above. Plasmid preparations were made using the Zippy Plasmid Mini Prep Kit (Zymo Research Ltd., Orange County, CA, USA). The horizontal electrophoresis of DNA was carried out using 1% Seakem LE agarose gels (Lonza, Basel, Switzerland). All cloning and molecular biology experiments were carried out according to established protocols, unless stated otherwise [35].

### 2.2. Plasmids Used in This Study

Plasmids pSG76-CS (GenBank: AF402780.1) [41], pST76-A (GenBank: Y09895.1), pST76-K (GenBank: Y09897.1), pSG76-K (GenBank: Y09894.1) [42], pZA31CFPtetR [43], pORTMAGE2 and pORTMAGE4 have been previously described [44]. Plasmid pSG78-A is identical to pSG76-A (GenBank: Y09892.1) except for containing an additional P-SceI site next to the I-SceI site. Plasmid pCDM4 was a kind gift from Prof. Mattheos Koffas [45]. Plasmid pUTLIQ_*vioABCDE* was a kind gift from Prof. Kanji Nakamura. Plasmids pKDsg-ack and pCas9cr4 were kind gifts from Prof. Kristala Prather [14]. All other plasmids were constructed for this study and are listed in Table 2. The process of their construction is described in Appendix B.

### 2.3. Rapid Electroporation of E. coli Cells

In our fast protocol to generate electrocompetent cells, *E. coli* cultures were grown in 10 mL of LB medium (harboring the appropriate antibiotic) to an OD600 value of 0.45–0.55. Cells were pelleted at 10,000 g for 2 min, and resuspended in 2 × 1.5 mL of ice-cold sterile Milli-Q water three times. The final resuspension took place in 40 μL of water. For electroporation, the cell suspension was mixed with the DNA, and the mixture was transferred into electroporation cuvettes harboring a 1 mm gap (Cell Projects Ltd., Herrietsham, UK). Electroporation was carried out in a MicroPulser electroporator (BioRad, Hercules, CA, USA) set to a voltage of 1.8 kV. The cells were recovered in 1 mL of LB, and shaken for 1–2 h at 30 or 37 °C (depending on the specific case). Finally, 10–100% of the recovery culture was plated on LB agar plates harboring the appropriate antibiotic.

### 2.4. Integration of Resistance Genes into Genomic IS Elements by Recombineering

For linear DNA-mediated recombineering, the PCR fragments targeting IS*1* were made by employing primers IS1ASpF and IS1ASpR, using the spectinomycin-resistant pCDM4 as a template, or IS1ACmR_F and IS1ACmR_R, using the chloramphenicol-resistant pSG76CS as a template. To test the effect of long homologies, primers IS1Sp100F and IS1Sp100R were used to amplify pCDM4. PCR fragments targeting IS*3* were made using primers IS3SpF and IS3SpR using pCDM4 as template. For primer sequences, see Appendix A.

A fully grown overnight culture of the host *E. coli* strain harboring pORTMAGE2 was diluted 100-fold into 50 mL LB + Ap. The culture was grown to an OD600 of 0.45–0.55, and the recombinase genes were then induced with a transient heat shock (42 °C, 15 min) followed by 10 min cooling on ice. This culture was used to make fast electrocompetent cells which were electroporated with 100 ng of linear PCR fragment (antibiotic resistance gene flanked by homology boxes corresponding to the respective IS element). After 2 h of recovery at 30 °C, 100 μL was spread on either an Sp or Km plate (depending on the resistance gene) and incubated overnight at 37 °C. The next day, single co-integrants were screened by colony-PCR using a pair of primers that hybridize to the resistance gene and the chromosomal segment neighboring the targeted IS elements, respectively.

In order to obtain double co-integrants, Cas9 selection was included. The protocol began as described above, but the recovery culture was grown overnight at 30 °C. The next day, the culture was diluted 50-fold in LB, grown to an OD600 of 0.45–0.55, used to make fast competent cells and 100 ng of pCas9_IS1 or pCas9_IS3 was electroporated. After a 2 h recovery in LB at 30 °C, Cm was added to select for the pCas9-derived plasmid and was shaken overnight at 30 °C. On the third day, 100 μL was spread on either Sp + Cm or Km + Cm plates (depending on resistance of the linear cassette) and incubated overnight at 37 °C. On the fourth day, double co-integrants were screened by colony PCR targeting all the potential genomic IS elements.

### 2.5. Integration of a Resistance Gene or the vioABCDE Operon into Genomic IS Elements Using NO-SCAR

A fully grown overnight culture of the host *E. coli* strain harboring pCas9-Cr4 and either pKDsg-IS1 or pKDsg-IS3 (targeting IS*1* and IS*3*, respectively) was diluted 100-fold into 10 mL LB + Cm + Sp. The culture was grown at 30 °C to an OD600 of 0.45–0.55 and 0.2% L-arabinose was added (inducing the λ-Red recombinases), followed by further growth at 30 °C for 15 min and cooling on ice for 10 min. This culture was used to prepare electrocompetent cells (see above) which were electroporated with 100 ng of linear PCR fragment (KmR gene flanked by homology boxes corresponding to the respective IS element). After 2 h of recovery at 30 °C, 900 μL was spread on a Km + aTc plate (to induce the CRISPR/Cas system) and incubated overnight at 37 °C. The next day, single co-integrants were screened by colony-PCR using a pair of primers that hybridize to the resistance gene and the chromosomal segment neighboring the targeted IS elements, respectively.

To integrate the *vioABCDE* operon into genomic IS elements, linear fragments containing the *vioABCDE* pathway and the resistance marker gene, flanked by homology boxes corresponding to the targeted IS elements, were generated by linearizing pST76AIS3::*vioABCDE*_KmR by SspI digestion or linearizing pST76AIS3::*vioABCDE*_SpR by KpnI+MunI+NheI co-digestion. The integration protocol was the same as above, except that 0.4% L-arabinose was used for the first induction, 600–800 ng of linear fragment was electroporated, and the recovery culture was spread on a Km+Sp+Cm+aTc plate and incubated overnight at 30 °C.

### 2.6. Integration of Non-Selectable Genes into Genomic IS Elements Using NO-SCAR

A fully grown overnight culture of the host *E. coli* strain harboring pCas9-Cr4 and either pKDsg-IS1 or pKDsg-IS3 (targeting IS1 and IS3, respectively), was diluted 100-fold into 10 mL LB + Cm + Sp. The culture was grown at 30 °C to an OD600 of 0.45–0.55 and 0.4% L-arabinose was added (inducing the λ-Red recombinases), followed by further growth at 30 °C for 15 min and cooling on ice for 10 min. This culture was used to prepare electrocompetent cells (see above) which were electroporated with 100 ng of linear PCR fragment (*gfp* gene flanked by homology boxes corresponding to the respective IS element). After 2 h of recovery at 30 °C, 0.5 μL was spread on a Cm + Sp + aTc plate (to induce the CRISPR/Cas system) and incubated overnight at 37 °C. The next day, single co-integrants were screened by colony-PCR using a pair of primers that hybridize to the resistance gene and the chromosomal segment neighboring the targeted IS elements, respectively.

### 2.7. Integration of Marked IS Elements into the Genome Using Transposition

A fully grown overnight culture of the host *E. coli* strain (MDS30 or MDS42) harboring the pSTtnp3tetR was diluted 100-fold into 5 mL LB + Cm + aTc, and was grown at 30 °C for 2 h. Then, the total 5 mL of this culture was used to make fast electrocompetent cells which were electroporated with 150–200 ng of pSG78A_full_IS3::SpR. After 1 h of recovery at 30 °C, 100 μL was spread on an Sp plate and incubated overnight at 37 °C. The next day, several colonies were replica-plated on Cm, Ap and Sp plates at 37 °C. One day later, Sp^R^Cm^S^Ap^S^ colonies (i.e., those that have lost both plasmids but retain the IS*3*::SpR) were inoculated into liquid cultures to generate glycerol stocks. The putative genomic insertions of IS*3*::SpR were localized using ST-PCR (see below).

### 2.8. Semi-Random Two-Step PCR (ST-PCR)

To localize a known sequence (IS*3*::SpR in our case) within a bacterial genome, we applied semi-random two-step PCR (ST-PCR) [46] with some modifications. Briefly, genomic DNA was PCR-amplified from the colonies of interest first using primer pairs Sp439Rev + CEKG2B and Sp110Fwd + CEKG2B, applying the following program: 94 °C, 2 min, followed by 6 cycles of 94 °C 30 s denaturation; 53 °C, 30 s, annealing with 1 °C decrease each cycle; and 72 °C, 3 min elongation. The DNA product generated by the first PCR was diluted 5x with TE buffer (10 mM Tris, 1 mM EDTA) and subsequently used as a template for nested PCR, with primer pairs SmFwd + CEKG4 and Sp347Fwd + CEKG4 corresponding to the first two PCRs, respectively. The applied program was 30 cycles of 94 °C, 30 s; 65 °C, 30 s; and 72 °C, 3 min. The products were separated on 1% agarose gels and bands of interest were extracted (using Viogene Gel/PCR DNA Isolation Kit) and sequenced with SmFwd or Sp347Fwd (depending on which was used for the nested PCR). Relative positions of the primers used for ST-PCR are shown in Appendix A.

### 2.9. Amplification of IS Elements Marked with Antibiotic Resistance Genes

A fresh overnight culture of the *E. coli* strain harboring the marked IS element was grown at 37 °C. Electrocompetent cells were made, transformed with the respective transposase-expressing plasmid (pSTinsAB’tetR or pSTtnp3tetR), spread on a Sp+Cm or Km+Cm plate and incubated overnight at 30 °C. A colony was picked and grown as an overnight starter culture in LB using the same antibiotics. The grown culture was diluted 10,000-fold and grown again at 30 °C in LB with antibiotics and the inducer, aTC (an uninduced control was grown in parallel). Dilution and growth were repeated after 24 h. The plasmids were then cured by growing the culture at 37 °C in LB without antibiotic selection for five days, applying a 10,000-fold dilution during each transfer. From the 5th fully grown serial culture, 1 μL was spread on 60× Sp gradient plate or 20× Km gradient plate and incubated at 37 °C for 24 h. The next day, 15 colonies from the high antibiotic-concentration area of the gradient plate were picked and replica-plated on Cm at 30 °C and Sp or Km at 37 °C. After overnight incubation, Cm^S^Sp^R^ or Cm^S^Km^R^ colonies were chosen and grown in liquid LB without selection at 37 °C. The fully grown cultures were saved as glycerol stocks and/or used for violacein quantification or to prepare genomic DNA for droplet-digital dPCR analysis. This process, defined as one round of transposase induction, was carried out up to three times.

### 2.10. Quantification of Violacein Production

To quantify the violacein production of liquid *E. coli* cultures, we applied the modified protocol of Zhu et al. [47]. The investigated strain was grown in LB with or without selection for 24 h at 37 °C. One milliliter of the culture was spun at 13,000 rpm for 10 min. After discarding the culture supernatant, 1 mL of DMSO (Molar Chemicals Kft., Halásztelek, Hungary) was added to the pellet. The solution was vigorously vortexed for 30 s to completely solubilize violacein and was centrifuged again at 13,000 rpm for 10 min to remove the cell debris. Then, 200 μL aliquots of the violacein-containing supernatant were transferred to a 96-well flat-bottomed microplate (Greiner Bio-One International, Kremsmünster, Austria), making three technical replicates, and the absorbance was recorded at 585 nm using a Synergy2 microplate reader (BioTek, Winooski, VT, USA).

### 2.11. Monitoring the Stability of Violacein Production

The stability of violacein production was assessed in two different experiments. The first type of experiment monitored the ratio of purple colonies in the lack of antibiotic selection. Overnight starter cultures of the investigated strains were grown in LB with selection (Sp or Km for the genomic co-integrants, Tc for pUTLIQ-carrying strains) at 37 °C. The culture was pelleted and resuspended twice in water to dispose of the antibiotics, and plated in appropriate dilutions on solid LB medium to obtain individual colonies. The number of purple colonies and total colonies was counted on each plate, and their ratio yielded the “day 0” value of cells expressing violacein. The cultures were then diluted 10,000-fold in LB medium and were fully grown without selection at 37 °C. Plating and enumerating the ratio of purple colonies was repeated to yield the ratio of cells expressing violacein for days 1–4. Each strain was investigated in three biological replicates.

The second type of experiment monitored the violacein production of liquid cultures grown in the lack of selection. Overnight starter cultures of the investigated strains were grown in LB with selection (Sp or Km for the genomic co-integrants, Tc for pUTLIQ-carrying strains) at 37 °C. The culture was then diluted 1000-fold in LB medium and was fully grown without selection at 37 °C. This dilution/growth cycle was repeated three more times. Each day, 1 mL of the fully grown culture was analyzed for the violacein content, as described above. Each strain was investigated in three biological replicates.

### 2.12. Droplet Digital PCR

To determine the copy numbers of marked IS elements within the bacterial chromsome, droplet digital PCR (ddPCR) experiments were performed using the EvaGreen protocol of BioRad QX200 Droplet DigitalPCR system (BioRad, Hercules, CA, USA). The template genomic DNA from *E. coli* strains were purified with the NucleoSpin Microbial DNA kit (Macherey Nagel), and digested with SspI (Thermo Fisher Scientific, Waltham, MA, USA). Final concentrations in reaction mixtures were the following: 1× QX200 EvaGreen Digital PCR Supermix (BioRad, Hercules, CA, USA), 3 pg digested *E. coli* genomic DNA, and 200 nM combined primer mix in 25 μL final volume. Twenty microliter reaction mixtures were used for droplet generation using the QX200 droplet generator. After partitioning, the samples were transferred into a 96-well plates, sealed and put in a T100 Thermal cycler (BioRad, Hercules, CA, USA). The following cycling protocol was used: 95 °C for 10 min, followed by 40 cycles of 94 °C for 30 s and 60 °C for 1 min followed by 5 min at 4 °C, 5 min at 95 °C, and finally at 4-degree infinite hold. The droplets were then read in the FAM channels and analyzed using the QX200 reader (Bio-Rad, Hercules, CA, USA). Primers specific for the Km-resistance gene (Km13fw and Km96rev) and the Sp-resistance gene (Sp347fw and Sp439rev) were designed using the ApE software (https://jorgensen.biology.utah.edu/wayned/ape, accessed on 17 January 2022) and ordered from MWG Eurofins Genomics Gmbh (Ebersberg, Germany). Final copy numbers were normalized to the bacterial single-copy *lacZ* gene, amplified with primers lacZ110fw and lacZ200rev. All primer sequences are listed in Appendix A.

### 2.13. Whole Genome Sequencing of Bacteria

For whole genome shotgun sequencing, a genomic library was prepared from four strains (B0, B1, B2 and B3) using the Nextera XT Library Preparation Kit (Illumina, San Diego, CA, USA) according to the manufacturer’s protocol. The libraries were sequenced with Illumina NextSeq 500 sequencers using 2 × 50 PE sequencing. Reads from all samples were aligned to the *E. coli* BLK09IS3::*vioABCDE*_SpR reference genome with BWA [48]. To find all insertion sites, we used Smith–Waterman algorithm [49] by filtering all reads where the IS*3* element’s 3′ 25 nt (TGATCCTACCCACGTAATATGGACA) or 5′ 25 nt (TGTCCACTATTGCTGGGTAAGATCA) sequence or their reverse complements can be found with a maximum 2 mismatches, and we then used the algorithm to select all those reads which could only be aligned with more than 10 mismatches to the +/− 200 nt region of the original single insertion site. Using the same algorithm, we trimmed all nt from each read which could be aligned to the insertion element. The resulting fastq file contained all reads marking the neighboring sequences of each insertion. This fastq file was remapped by BWA to the *E. coli* BLK09IS3::*vioABCDE*_SpR reference genome. To mark independent insertion sites, we searched for peak positions different from those of sample B0. To calculate the overall number of independent insertions and repeated re-insertions, relative coverage was measured in the complete alignments by calculating the ratio of reads at +/− 10 nts from peak borders.

## 3. Results

The general strategy tested in this work was to use ISes as landing pads to integrate transgenes into bacterial genomes, followed by their copy number amplification using transposition. There are several considerations supporting this idea: (i) to the best of our knowledge, ISes have never been found to be essential; (ii) ISes are near-ubiquitous, having been detected in 76% of bacterial genomes [50]; (iii) the type, number and exact position of ISes in each sequenced genome are well known; (iv) copy-and-paste types of ISes have been identified; and (v) the mobility (transposition rate) of various IS types is often known, allowing the anticipation of potential transgene amplification. We chose to test two copy-and-paste type of IS elements, IS*1* and IS*3*, which are at the high-end and low-end of the transpositional activity scale in *E. coli*, respectively [51]. Concerning the copy numbers of these elements, wild-type (wt) *E. coli* K-12 MG1655 carries 8 copies of IS*1* and 5 copies of IS*3*, while *E. coli* BL21(DE3)pLysGold carries 28 and 4 copies, respectively. In the course of an earlier project carried out in our laboratory, however, we deleted all ISes from MG1655 [38], and saved intermediate strains carrying 2 or 1 copies of IS*1* (called MDS39R2 and MDS42IS1, respectively), or 3, 2 or 1 copies of IS*3* (called MDS16, MDS27 and MDS30, respectively). Similarly, during the course of BL21(DE3) genome reduction [40], we generated BLK09 which carries two active copies of IS*3*, and BLK16, where the same two copies of IS*3* are inactivated by nonsense mutations within the transposase gene. We were therefore in the fortunate situation that we could choose multi-deletion strains carrying limited copies of the targeted elements, thereby simplifying the analysis of transgene integration.

### 3.1. Targeting ISes by Recombineering

In the first type of experiment, we sought to establish whether targeting IS*1* and IS*3* were just as straightforward as targeting other nonessential genes of *E. coli*. We generated linear dsDNA fragments by PCR that harbored an antibiotic resistance gene flanked by two appropriate homology boxes (provided by the PCR primers). Note that the linear fragments did not harbor the inverted repeats of the IS to avoid the binding of the transposase enzyme and potentially causing fragment entry into the genome by random transposition. We expressed the λ-Red recombinases from the inducible pORTMAGE2 to facilitate the double-crossover between the respective chromosomal IS element and the resistance-cassette. In parallel experiments, we targeted the single IS*1* copy of MDS42IS1 and the single IS*3* copy of MDS30. As expected, the colonies obtained on selective plates were genomic co-integrants: using colony PCR, 10 out of 10 colonies tested positive in both the IS*1*-targeting and the IS*3*-targeting experiments. Therefore, integrating selective markers into IS elements by recombineering is not different to targeting other genes. It is noteworthy, however, that using SpR as a selection marker elevated the yield of recombinants more than 20-fold compared to using CmR (15.26 colony/ng DNA vs. 0.718 colony/ng DNA, respectively, as listed in Appendix A).

In the second class of experiments, we repeated the recombineering process as above, but used strains carrying two copies of the targeted ISes: MDS39R2 for IS*1* and BLK09 or BLK16 for IS*3*. All colonies obtained on the selective plates were PCR-screened for integration at both loci. On the one hand, when targeting IS*1* in MDS39R2, only 13 out of 25 colonies (52%) carried the SpR in either of the IS*1* elements; for the remaining 12, the SpR cassette was integrated at an unknown locus. The majority of the known integrations occurred at the YeaJ::IS*1*, the minority at the ais::IS*1* loci (11 vs. 2, respectively) (see Appendix A for an example). On the other hand, when targeting IS*3* in BLK09 and BLK16, the resistance cassette was always detectable at either one of the two loci. Again, there was a bias favoring the integration into the IS*3* at locus 1 (gene ECBD2567) vs. locus 2 (gene ECBD0875) (7:3, respectively). The ratios of integration at each locus are shown in Figure 1. Double co-integrants were never obtained this way in either the IS*1* or the IS*3*-targeting experiments. When targeting the two copies of IS*1*, we confirmed the observation that using SpR as a transgene yields more co-integrants than using CmR (16.73 vs. 0.33 colony/ng, respectively, see Appendix A), prompting us to use SmR in further experiments.

In our third group of experiments, we attempted the integration of the 7.3 kbp-long *vioABCDE* cassette into the bacterial genome using λ-Red-mediated recombineering. The *vioABCDE* operon originates from *Chromobacterium violaceum*, and is responsible for producing the purple pigment violacein [52]. Since generating linear *vioABCDE* fragments (flanked by appropriate homologies) using PCR was unsuccessful, we cloned the operon into the temperature-sensitive pST76-A vector, flanked by homology arms corresponding to IS*1* or IS*3* elements, making pST76AIS1X::*vioABCDE* and pST76AIS3X::*vioABCDE*, respectively. (Note that neither of these plasmids contains the inverted repeats of the targeted IS.) We then went on to include an antibiotic resistance gene (SpR or KmR) inside the cassette next to the *vio* operon, within the segment flanked by IS-specific homologies. Four such plasmids were generated: pST76AIS1X::*vioABCDE*_SpR; pST76AIS1X::*vioABCDE*_KmR; pST76AIS3X::*vioABCDE*_SpR; and pST76AIS3X::*vioABCDE*_KmR. Linear DNA cassettes for recombineering were generated from these plasmids by restriction digestion. When electroporating these cassettes, we obtained mixed results: on the one hand, if targeting genomic IS*1* in MDS42IS1, correct recombination events could be verified, but none of the co-integrant colonies were purple. This was the case that despite the fact that all four plasmids used to generate the cassettes granted their host a dark purple colony phenotype. On the other hand, when targeting the IS*3* of MDS30, the PCR-verified genomic co-integrants displayed a purple color: three out of nine colonies (33%) were PCR-positive when using SpR; and 9/10 (90%) were positive when using KmR as a selection marker (Appendix A).

Targeting *E. coli* strains with two copies of IS*3* within the genome also provided the integration of the *vioABCDE*_SpR cassette with rates acceptable for practical purposes: the fractions of colonies harboring single co-integrants in *E. coli* BLK09, BLK16 and MDS27 strains fell between 20 and 75% (Table 3).

Double co-integrants were never obtained this way. We nevertheless demonstrated that the 9074 bp- and 9359 bp-long *vioABCDE*-SpR and *vioABCDE*-KmR cassettes, respectively, could be targeted at genomic IS*3* elements by recombineering. In both experiments targeting IS*3* of BLK09 or BLK16, the majority (>90%) of the colonies displayed a purple color, indicating a relatively low rate of false positive resistance. In strain MDS27, however, <50% of colonies were purple to the naked eye, and some of the white colonies were true co-integrants verified by PCR. This exemplifies the strain-dependence of the ratio of colonies with a functional expression of the inserted operon. The bias of the integration seen for resistance genes was also present when inserting the five-gene operon, although to a variable extent (Figure 2).

When targeting *E. coli* MDS16, harboring three IS*3* elements, we managed to detect single co-integrants at all three loci, although we only found one insertion at b0374::IS*3* out of the 20 colonies tested (Figure 3). We therefore showed that a relatively long (>9 kbp) operon can be routinely inserted in a single copy into *E. coli* strains having one, two or three IS elements within their genomes, relying solely on recombineering. We have no reason to think that IS elements could not be used as recombination targets if they were present at higher copy numbers on the chromosome, as only the localization of the transgene insertion becomes more tedious. The absolute efficiencies of obtaining true co-integrants are shown in Appendix A.

### 3.2. Targeting ISes by CRISPR-Mediated Recombineering

After demonstrating the utility of recombineering to load genomic ISes with selectable transgenes, we continued with testing the process of CRISPR/Cas-mediated recombineering transgenes into genomic IS elements. Among our objectives were to increase both the efficiency of recombineering and the maximum size of the insertable payload, as well as attempt the insertion of unselectable genes and the insertion of a transgene into multiple copies. Our CRISPR/Cas-mediated experiments could be divided into two strategies: on the one hand, we used subsequent CRISPR/Cas cleavage after the recombineering step as an additional tool of counterselection against the wild-type (wt) genotype. In these experiments, the λ-Red recombinase enzymes and the CRISPR/Cas machinery were provided by pORTMAGE2 and pCas9, respectively. On the other hand, we also tested concomitant CRISPR/Cas cleavage during the recombineering step to aid the recombination process by generating free DNA ends (in addition to the selective effect mentioned previously). In this latter case, we used the NO-SCAR system (i.e., pKDsg and pCas9cr4), previously developed by the Prather laboratory for a similar purpose [14].

When targeting a single copy of IS*1* or IS*3* in *E. coli* MDS42IS1 or MDS30, respectively, subsequent CRISPR/Cas cleavage (with pCas9IS1 or pCas9IS3) was occasionally able to increase the absolute efficiency of recombination, but this effect was inconsistent, with certain cases even falling short of the recombination rates seen in simple recombineering (Appendix A: two cases improve, two cases worsen SpR integration into MDS42IS1). However, when it came to targeting strains harboring two copies of the respective IS element, the advantage of subsequent CRISPR/Cas cleavage became readily apparent: double co-integrants were routinely obtained using multiple protocols. In the end, we simplified the process to two transformations (described in the Methods): the linear DNA is transformed first for recombineering, and the appropriate pCas9 plasmid cleaving the wt form of the IS element is transformed the next day to enforce selection. The pCas9IS1-mediated counterselection in MDS39R2 resulted in 1/7 (14%) or 3/10 (30%) of the tested colonies harboring the resistance cassette at both loci of IS*1* (for short and long homologies, respectively). Counterselection using pCas9IS3 was even better with 15/20 (75%) of the colonies proving to harbor double co-integrants in BLK16. Double IS*3* co-integrants were also successfully obtained in MDS27, displaying the portable nature of this technique. Figure 4 shows the fraction of colonies harboring double co-integrants after one or two cycles of recombineering, using either short or long homologies in MDS39R2.

In addition to generating double co-integrants, the great value of applying CRISPR/Cas-based facilitation was unveiled upon the integration of unselectable genes into the chromosome. Peculiarly, we managed to insert the *gfp* gene into the IS*3* of *E. coli* MDS30 using recombineering and subsequent CRISPR/Cas selection. In the course of these experiments, we demonstrated that a 20 nt spacer is equal or superior to a 30 nt spacer in pCas9IS3, concerning efficiency, and up to 20% (2/10) of colonies obtained proved to be positive by PCR. The integration of *gfp* into the IS*1* element of MDS42IS1 was also successfully achieved using recombineering and a concomitant CRISPR/Cas cleavage provided by the NO-SCAR system (pKDsg-IS1 and pCas9cr4). In the latter experiments, up to 10% of the colonies (3/30) were found to be PCR-positive. The elevated green fluorescence levels of both engineered strains were verified using a microplate reader (Appendix A).

We then tested the effect of CRISPR/Cas cleavage on the integration efficiency of long (>9 kbp) selectable DNA cassettes. Since integrating our *vio* operon into IS*1* targets by recombineering led to the loss of violacein production (see above), we focused on experiments targeting IS*3*. First, we tested the effect of subsequent Cas cleavage using pCas9IS3. We found that in most of the experiments targeting MDS27, BLK09 or BLK16 (using either SpR or KmR as selection markers), subsequent Cas cleavage not only failed to improve the efficiency of pORTMAGE-mediated recombineering, but on many occasions, the recombinant cells were missing altogether. In a few cases, nevertheless, we did manage to obtain single *vioABCDE*-KmR insertions into either of the IS*3* elements of BLK09 (Figure 5 and Appendix A) or BLK16 but double co-integrants were not obtained this way. Second, we tested the effect of a concomitant Cas cleavage using the NO-SCAR system. We started by inserting the *vioABCDE*-KmR cassette into the single IS*3* of MDS30 using NO-SCAR (pKDsg-IS3 and pCas9cr4). We obtained single co-integrants with a low absolute yield (on the order of 10^−2^ correct colonies/ng), the high ratio of correct colonies (6/10) nevertheless permitted the easy detection of true recombinants. The same system was used to target the two copies of IS*3* residing in MDS27, with 7.7% of colonies displaying the operon inserted at both loci. The absolute efficiencies of obtaining recombinants using CRISPR/Cas-assisted recombineering are listed in Appendix A.

### 3.3. Amplification of Cargo Genes Using Copy-Paste Transposition of IS Elements

After verifying whether genes and large operons can be integrated into IS elements, our next goal was to achieve the copy number amplification of these co-integrants via the process of copy/paste transposition. Our initial experiments dealt with the amplification of single resistance genes (e.g., SpR) inserted into IS*1* or IS*3*. The respective transposase genes were expressed from a plasmid in trans, and host cells carrying copy number-amplified resistance genes were selected on antibiotic-gradient plates. Colonies picked from the high-end of the gradient plates went through genomic DNA preparation and ddPCR to quantify the copy number of the resistance gene. These pilot experiments revealed two important conclusions. First, both IS*1* and IS*3* could be used to yield highly resistant colonies upon transposase induction (Appendix A). IS*1* experiments however, tended to be less reproducible, with unknown factors influencing the observable increase in resistance (Appendix A). Second, the ddPCR of the initial amplification experiments indicated a major increase in the copy numbers of the resistance gene, increasing to 20 copies/cell in the first round (Appendix A, left). However, when repeating the ddPCR on the same cells after curing them from the transposase-expressing plasmids, the detectable copy numbers fell to the range of 2–4 (Appendix A, right). This indicated that most of the high resistance was caused by the transposition of the resistance cassette into the transposase-expressing plasmid. We therefore engineered temperature-sensitive transposase plasmids (pSTinsAB’tetR and pSTtnp3tetR for IS*1* and IS*3*, respectively), and plated the cells on gradient plates only after the plasmids were cured from the cells (as described in the Methods section). This guaranteed that only chromosomal co-integrants of the resistance gene were selected after transposase expression.

Due to the low reproducibility of IS*1*-mediated amplification and to the fact that violacein expression was not observable upon IS*1*-targeted integration (described in the previous section), we primarily focused on the amplification of the violacein operon (*vioABCDE*) using IS*3* elements. Strains of *E. coli* BLK16_IS*3*::*vioABCDE*_SpR, generated by recombineering (see above), were used as starting points. These carried a single copy of the violacein operon inside the IS*3* element residing at locus 1, along with an SpR marker. After the first round of transposase expression, we analyzed five colonies by ddPCR (targeting the resistance marker) from the high-end of the gradient plates and found the mean copy number to be 5.50 (±1.22) (Figure 6). Choosing a colony displaying 7.59 copies and re-expressing the IS*3* transposase led to a further significant elevation of the mean copy number within colonies displaying high resistance levels (8.03 ± 1.21, *n* = 4) (Figure 6). Again, we chose the colony displaying the highest copy number (9.10) and expressed the IS*3* transposase for the third time. The mean copy number of the colonies analyzed in the third round did not significantly differ from those of the second round (Figure 6). However, by “cherry picking”, we were still able to find individual clones displaying further elevated copy numbers (see below).

We then sent a series of four strains obtained from the above experiment (the starting strain and one clone from each round of transposase induction) for whole genome sequencing. Our aims were (i) to verify the copy numbers of the amplified IS*3*::*vioABCDE*_SpR elements; (ii) to analyze the fraction of mutant or truncated *vioABCDE* operons; (iii) to test whether amplification leads to tandem repeats or a random scatter of the loaded IS*3* element within the genome; and (iv) to check for unexpected genomic rearrangements. The sequence of the starting strain (B0) confirmed the presence of the *vioABCDE* operon, along with the SpR marker within the IRs of the IS*3* element residing at locus 1 (genomic coordinates 649,839–659,074). The sequence of strain B1, obtained in the first round of transposase induction displayed a 5.71-fold elevated relative coverage of the loaded IS*3*, and unexpectedly, of a 14 kbp segment directly downstream of the IS (amplified region: 649,839–673,907). This stands in good agreement with the 5.59 copy numbers measured by ddPCR. The sequencing of strain B2, which is not a direct descendant of B1, revealed a 11.32-fold increased relative coverage of a similarly large genomic segment delimited on the left side by the left IR of the loaded IS3 (coordinates: 649,839–677,086). Interestingly, it also displayed a further 5.44 copies (totaling to 16.76) of an interior segment (659,074–667,565), which is directly downstream of the manipulated IS. The approximately 11-fold increase in the copy number of the violacein operon roughly supports the 9.1 copies indicated by the ddPCR. Finally, strain B3, a descendant of B2, was expected to harbor 26.11 copies of the resistance marker, based on ddPCR. Sequencing did not confirm this, as the relative coverage of the genomic region encompassing the loaded IS*3* (649,839–677,086) nevertheless increased from 11.32 to 12.51, and the number of additional copies of the interior segment downstream of the IS (659,074–667,565) decreased from 5.44 to 3.54. We therefore concluded that although the third round of induction did not increase the mean copy number of the inserted operon, screening a low number of colonies (<10) permitted the identification of a clone that displayed further amplification of the transgenes. We note that in another clone of the same series of induced strains, we detected 9.89 copies, and in another induction series performed in parallel, we measured 10.71 and 11.83 copies of SpR using ddPCR. We also experienced that in a small fraction of analyses (1–5%), the ddPCR results were misleading (data not shown), underlining the importance of result verification. The sequenced strains did not harbor the loaded IS*3* scattered throughout the genome, but rather amplified in tandem repeats. However, the IS copies did not form a back-to-back array since, rather unexpectedly, they were amplified together with a large (14–18 kbp) DNA segment lying directly downstream to them.

The violacein content of the starting strain and its derivatives obtained after various rounds of transposase induction were routinely measured. The amount of violacein released from the BLK16IS*3*::*vioABCDE*_SpR derivatives displayed a good correlation (R^2^ = 0.98) with the copy number of the *vioABCDE* operon inferred from its relative sequencing coverage (Appendix A). Copy numbers of the SpR marker determined by ddPCR, relative violacein content and the relative sequence coverage of the violacein operon (acquired by whole genome sequencing) are displayed in Table 4 for each sequenced strain of the IS*3*::*vioABCDE*_SpR-amplification process performed in BLK16.

To verify the general applicability of the inPOSE strategy, we tested the amplification using IS*3* elements residing elsewhere in the same strain, employing further strains of *E. coli*, as well as using a different selection marker. Inducing the IS*3*::*vioABCDE*_SpR at locus 2 in strain BLK16 for one round also yielded purple colonies with significantly elevated copy numbers of the operon (3.28 ± 0.15) (Appendix A). Importantly, when analyzing white colonies by ddPCR, the mean copy number was even higher (8.20 ± 2.79). In this experiment, we induced a white colony for two further rounds, and identified white colonies harboring ever higher copy numbers of the SpR marker (31.70 and 49.14 in round 2 and 3, respectively); the increase in the means, however, was not significant (Appendix A). One round of transposase induction also worked in BLK09IS3::*vioABCDE*_SpR (locus 2) (Figure 7), and the obtained mean copy number was 3.88 (±1.68). Interestingly, using KmR in the same locus, we only found a maximum copy number of 2.27, with the mean not being significantly elevated (1.51 ± 0.52).

To test a completely different genetic background, amplification was also tested in strains MDS27 and MDS30, derivatives of *E. coli* K-12. For *E. coli* MDS30IS*3*::*vioABCDE*_KmR, one round of transposase induction boosted the number of highly resistant colonies compared to the uninduced ones (Appendix A). The mean copy number of the KmR marker was also significantly elevated (7.57 ± 2.39); however, only a small fraction of the colonies growing on the plate were purple. For *E. coli* MDS27doubleIS3::*vioABCDE*_KmR, a strain that had both copies of IS*3* loaded with *vioABCDE*_KmR at the starting point, we carried out three rounds of induction. Similarly to that seen for MDS30IS*3*::*vioABCDE*_KmR, the first round of induction elevated the copies of KmR marker to 4.78 (±0.76), but most of the colonies were white (Appendix A). The second and third rounds of induction, both of which were carried out choosing a purple colony, could not significantly increase the mean copy number of the marker (5.7 ± 2.66 and 4.0 ± 0.64, respectively). Again by cherry picking, we could find colonies carrying up to 7.8 copies. Importantly, the second and third rounds of this experiment revealed that in the lack of induction, the majority of the colonies remain purple (Appendix A), but the obtainable mean copy numbers are somewhat limited (4.07 ± 1.96 and 2.95 ± 1.16, respectively).

### 3.4. Stability of Genomic Co-Integrants

A key phenotypic trait of engineered strains is the stability of expressing the inserted transgenes in the lack of selection. We compared strains harboring chromosomal *vioABCDE* operons to those carrying the same operon on multi-copy plasmids using two different tests. Our first test monitored the fraction of cells displaying violacein production. Our controls were *E. coli* BL21(DE3), DH5α and MG1655, all transformed with pUTLIQ_*vioABCDE*. From our collection of genome-engineered strains, we tested three versions of BLK16IS*3*::*vioABCDE*_SpR (carrying 1, 5.59 or 10.5 copies of IS*3*::*vioABCDE*_SpR, respectively), and strain MDS27IS*3*::*vioABCDE*_KmR, (carrying 5.8 copies). All three control strains displayed a steep decrease in the ratio of purple colonies, and completely lost the purple phenotype by generation 40. On the contrary, all four genomic co-integrants tested in this experiment retained purple color in >96% of their colonies, with the single copy strain not displaying loss of function at all (Figure 8). We note that (i) the multi-deletion strains (MDS42 and BLK16) of *E. coli* displayed complete plasmid loss even faster than their conventional, non-reduced counterparts; and (ii) further single and double copy-harboring strains (BLK16IS*3*::*vioABCDE*_SpR (1 copy, locus 2) and MDS27IS*3*::*vioABCDE*_KmR, (2 copies)) did not display any loss of function either (Appendix A). We therefore conclude that our genomic co-integrants can be used for the prolonged expression of transgene arrays in the lack of selection with a negligible fraction of cells displaying complete loss of production.

Our second test assaying the stability of violacein production in the lack of selection measured the levels of violacein extracted from the cultures each day. As expected, we saw the rapid decline of violacein levels in the case of *E. coli* BL21(DE3), MG1655 and MDS42 strains initially harboring pUTLIQ_*vioABCDE* (Figure 9). *E. coli* DH5α displayed a somewhat smaller, but still significant loss of 35% (*p* = 0.012 with a two-tailed, unpaired *t*-test). In contrast, no significant decrease in the violacein levels were observed on day 4 (compared to day 1) for strains BLK16IS*3*::*vioABCDE*_SpR carrying 5.59, 9.1 and 10.5 copies of the violacein operon, respectively, and strains MDS27IS*3*::*vioABCDE*_KmR carrying 2 and 5.8 copies, respectively. Furthermore, in this test, the violacein levels produced by MDS27IS*3*::*vioABCDE*_KmR carrying 5.8 copies significantly surpassed that of all other strains, including the four strains carrying pUTLIQ_*vioABCDE* (Figure 9 and Appendix A).

### 3.5. Transposition of Marked IS3 Elements into the E. coli Chromosome

Finally, we asked whether IS*3* can be transferred into a bacterial strain of choice by controlled transposition. This process, which can be referred to as the potential “Step 0”of the inPOSE protocol (Figure 10), could be valuable for future users who wish to utilize IS*3* for chromosomal cloning and amplification inside a strain that has no copies of this element at all. For the process of IS*3* entry into the chromosome, we decided to use transposition, catalyzed by the plasmid-encoded transposase used in the transgene copy-amplification experiments above. To make the IS*3* entry selectable, we used the IS*3*::SpR allele generated in the course of SpR recombineering in MDS30. The IS*3*::SpR cassette was PCR-amplified from the chromosome and cloned into a suicide plasmid that was unable to replicate in strains lacking the *pir* gene. This donor plasmid, pSG78A_full_IS*3*::SpR, was transformed into *E. coli* MDS42 carrying the transposase-expressing pSTtnp3tetR, followed by plating on Sp plates. Replica plating revealed that 10% (2/20) of the obtained colonies were Sp^R^Cm^S^Ap^S^, indicating that they had lost both the donor plasmid and the transposase plasmid, but retained the SpR gene, presumably due to its transposition into the genome. To verify transposition and identify the locus of entry, we carried out ST-PCR on the chromosomal DNA prepared from two colonies. Two pairs of PCR reactions were carried out, as described in the Methods (Appendix A). Sequencing a unique product (from SmFwd + CEKG4 PCR on colony 11) confirmed IS*3*::SpR insertion into the *pqiA* gene at position 1,012,519 (using NC000913.3 coordinates). In a similar experiment, the transposition of IS*3*::SpR into MDS30 was induced and selected, and 33% (10/30) of the colonies displayed a Sp^R^Cm^S^Ap^S^ phenotype. Most of ST-PCR reactions carried out on the genomic DNA of ten such colonies generated unique PCR patterns, probably indicating different points of insertion (Appendix A). Sequencing a PCR product from one of the colonies revealed IS*3*::SpR integration into the 5′ untranslated region of the *waaL* gene (position 3,796,945, NC000913.3 coordinates). We therefore demonstrated the transposition-mediated entry of IS*3*::SpR into two different strains of *E. coli* (one IS-free, the other harboring a single copy of IS*3*), with IS*3*::SpR entering the chromosome at two distinct loci.

## 4. Discussion

The interest in techniques capable of rapidly integrating genetic constructs into the genome has been present for decades. As discussed in the introduction, multiple solutions have been devised for this problem, but only a few are capable of cloning transgenes in multiple copies within the chromosome. In addition, these methods require the serial transfer of individual insertions into an accumulator strain [19], or are limited by the number of different phage attachments sites available in the genome [20]. The pre-integration of further attachment sites into the chromosome would also be an inconvenient solution due to its tedious nature. To overcome these limitations, we developed a method that uses mobile elements as landing pads, i.e., recombination target sites. The recombination process is mediated by the λ-Red recombinase enzymes expressed from a plasmid, providing portability to our method. The transposase activity is used for the copy number amplification of the inserted transgene(s) in the second stage of our process.

Recombineering single selectable transgenes into IS elements achieved yields of recombinants mostly in the range of 0.5–20 recombinants/ng DNA, which meets or exceeds the efficiency values reported in the literature [13,53] and is sufficient to conveniently obtain co-integrants in a single try. If one wishes to generate chromosomal libraries, the use of longer homologies could be a viable solution; in our experiments, increasing homology length 2.5-fold (to 100 bp) elevated the yield of recombinants over 13-fold to up to 268 recombinants/ng DNA (Appendix A). Despite low expectations based on earlier reports on the recombineering of >9 kbp-long DNA fragments into the genome [13], we readily obtained single co-integrants of the marked *vioABCDE* operon in the chromosomes of strains carrying one, two or three copies of IS*3*. Although our PCR screens also suggested the formation of single co-integrants when targeting IS*1* elements, the lack of *vioABCDE* function prevented us to use these strains further. We suspected a technical fault during the generation of the linear fragments as the reason; this, however, was not investigated further. When inserting a single resistance gene into strains with a single copy of the targeted IS, all obtained colonies carried the transgene within that IS. When two IS*3* copies were present in the genome, still all integrations took place at known loci. In all other cases, however, a certain fraction of the obtained colonies did not harbor the transgene(s) in any of the IS copies, indicating integrations at unknown loci. These events happened more often when targeting IS*1* or when attempting to insert a long DNA cassette, possibly resulting from a higher rate of illegitimate recombination.

Then, we complemented recombineering with CRISPR/Cas cleavage to: (i) increase the rate of recombination by providing free DNA ends, if used concomitantly [54,55]; (ii) increase the maximum size of the payload that is insertable [13]; (iii) allow the insertion of unselectable genes; and (iv) allow the insertion of transgenes into multiple copies in a single step. The use of CRISPR/Cas to aid the selection of recombinants or to facilitate the recombination process did not result in the increase in recombination efficiency when targeting a single IS in the genome. However, obtaining double co-integrants was only possible with CRISPR/Cas selection. This is not surprising, for as long as an unloaded (wt) IS remains in the genome, the Cas9 ribonucleoprotein will find and cleave it, and introducing a double-strand break into the chromosome is a very powerful form of negative selection in *E. coli*. We did not manage to obtain triple co-integrants, however, when targeting the three copies of IS*3* found in MDS16. This could indicate that the frequency of triple recombination is lower than the frequency of double-strand break-repair, only allowing the formation of double recombinants. Perhaps transforming the linear DNA in multiple (≥3) cycles prior to CRISPR/Cas selection could solve this issue, this, however, was not attempted. In addition to obtaining double co-integrants, the use of CRISPR/Cas was also essential in the process of recombining an otherwise unselectable gene (*gfp*) into the chromosome. Despite our attempts, we were not able to obtain double *gfp* integration into the genome, possibly indicating that applying multiple mechanisms of selection (i.e., resistance and Cas cleavage) has a synergistic effect on the formation of recombinants. The use of CRISPR/Cas cleavage concomitantly or subsequently to the recombineering step did not show noteworthy differences when inserting single genes into the genome. However, integrating the >9 kbp DNA cassette in multiple copies into the *E. coli* chromosome was only possible when using the NO-SCAR system. We attribute this to the facilitating effect of free DNA ends (generated by the concomitant cleavage) on recombination efficiency.

The second phase of our newly developed technique was aimed at the amplification of the inserted cargo. Earlier techniques, mentioned in the Introduction, achieved this goal by serially passaging cells in increasing antibiotic or biocide concentrations to select for RecA- or Cre-mediated gene duplications [21,22,23]. Our strategy is related; however, it uses the plasmid-expressed transposase enzyme corresponding to the targeted IS for the amplification process. It does not require an increasing selection pressure to evolve the multi-copy harboring strain; the elevated antibiotic levels present on a gradient plate are only required for the “readout” of the transposition process.

In our gene-amplification experiments, the induction of the transposase had a marked effect on the distribution of resistance levels within the population in the first round (Appendix A). In the second and third rounds of induction, however, the difference in the distribution of resistance levels between the induced and uninduced population was smaller, if noticeable (Appendix A). This may indicate that at higher starting copy numbers of the marked IS, even the leaking expression of the transposase was sufficient to facilitate sufficient transpositions to generate highly resistant cells. As an alternative explanation, the spontaneous RecA-mediated duplication of marked ISes most likely takes place at all copy numbers; the duplication of many copies (at later rounds), however, probably has a more dramatic effect on resistance than the duplication of a single copy (in the first round). Either way, these spontaneously arising clones of elevated resistance, even if arising less frequently, can harbor just as many copies as the induced clones.

A key parameter of transgene-carrying strains is the stability of transgene expression, especially in the absence of antibiotic selection. Our first experiment measuring this parameter declared colonies as producers and non-producers in a binary fashion, based on their purple or white color, respectively. This experiment indicated a rapid loss of the producer phenotype for plasmid-carrying strains, but the maintenance of production by >96% of the cells in the case of genomic co-integrants (Figure 8). In a more quantitative analysis, we measured the violacein levels extractable from liquid cultures grown in the lack of selection. Again, the significant decrease in violacein levels was only seen in the case of plasmid-based expression, and not in the case of chromosomal expression (Figure 9). The strong strain dependence of violacein production was also apparent in the second type of experiment, both for plasmid-based and chromosomal transgene expression.

The utility of the inPOSE system for transgene entry by homologous recombination, and transgene amplification by copy/paste transposition was demonstrated in the experiments above. We have no reason to think that IS*3* is the only element applicable for this purpose. Nevertheless, for those who wish to use IS*3* for inPOSE, but their targeted bacterial cell lacks this element, we showcase a transposition-based method to insert a marked copy of IS*3* into the genome of the chosen host bacterium. We refer to this as “Step 0” of the inPOSE protocol, which can be readily continued with Step 1 if using a different antibiotic for transgene insertion. The entire inPOSE process is graphically displayed in Figure 10, and the condensed protocol is provided as a list of tasks in Appendix C.

The potential utility of inPOSE is easy to envision: numerous examples are available in the literature requiring the expression of RNA- or protein-encoding genes from the bacterial chromosome. For example, *E. coli* strains carrying an elevated number of rRNA operons [56] or extra copies of rare tRNA genes [57] have been engineered for fundamental and applied scientific purposes, respectively. A more common need is to express heterologous enzymes or entire enzymatic pathways from genes inserted into the bacterial chromosome [22,23,58,59]. A number of such strains published to date displayed improved genetic stability even in the lack of chemical selection, a crucial feature for the fermentation industry [22,23]. The improvement of production titers by copy number amplification has also been demonstrated [22,23]. In certain cases, the proper balancing of the heterologous pathway was granted by the isolated alteration of certain enzyme-encoding gene’s copy number [24,58,59]. The inPOSE protocol fits into the line of such experiments, providing a flexible tool for the integration and amplification of heterologous pathway-encoding gene constructs. Ultimately, the use of multiple IS elements within one engineered strain could allow a combinatorial alteration of the copy numbers of genes constituting the pathway, allowing the selection for the best producer using a high-throughput assay.

Despite the multiple advantages of using inPOSE for chromosomal gene cloning and amplification, certain challenges are still apparent which will require further research. For example, the suboptimal performance of IS*1*-mediated integration and amplification in the course of our work remains unexplained, and we therefore cannot predict which further IS elements will be optimal choices for this type of task. The systematic testing of ISes could provide us with more insight concerning the nature of the problem. Another interesting finding was that the transposition of the loaded ISes did not result in a random scattering of the IS within the chromosome, but in the formation of tandem repeats instead, along with the amplification of the 14–18 kbp-long genomic segment lying directly downstream of the IS. This is probably the result of the transposase enzyme not acting on the right IR of the IS, but instead on a pseudo-terminus, lying further away. Transposases acting on the pseudo-termini of transposons have been described [60,61]; the question here is why the transposase misses the canonical right terminus of the loaded IS*3*. The reason may be that the 827 bp missing from the center of the interrupted IS*3* removes an unknown protein or RNA factor necessary for correct transposition, and the transposase gene expressed in trans fails to complement this missing function due to its codon-optimized nature. We will try to identify this factor and provide it in vivo to obtain controlled transposition that is equivalent to the natural process. Furthermore, the expression levels of the integrated transgenes and operons were in certain cases unexpected, or hard to predict. A notorious example can be seen in Figure 9: MDS27IS*3*::*vioABCDE*_KmR strains carrying 5.8 copies of the KmR gene (measured by ddPCR of the KmR gene) produces much more violacein than the BLK16IS*3*::*vioABCDE*_SpR strain carrying 9.1 or 10.5 copies. On the contrary, BLK16IS*3*::*vioABCDE*_SpR strains displayed violacein production that was more or less proportional to the copy number of the operon, as apparent in Figure 9 and Table 4. This obviously points to the strain dependence of violacein production, and underlines the general necessity of phenotypical testing, in addition to analyzing copy numbers by genetic methods. Other possible causes of unexpected production levels are the fact that ddPCR measures the resistance gene, not the presence of the complete, correct operon. When considering long-term stability of genomic co-integrants, one should also take into account the occasional loss of tandem repeats by homologous recombination. The initiation of this process is probably the cause of the slight decrease seen in Figure 8 for the multi-copy insert carrying strains after 25–40 generations of growth. We anticipate that the recombination-mediated loss of operon copies can be avoided by re-engineering the transposition process, as mentioned above. An alternative solution could be the removal of the *recA* gene from the genome. If transposition was seriously impaired, the *recA* inactivation could be the final step of the amplification protocol [21].

## 5. Conclusions

We showed the successful genome editing of *E. coli* by homologous recombination targeting IS*1* or IS*3* elements, using CmR, SpR or KmR as selection markers. We demonstrated how to solely integrate a resistance gene or a five-gene operon marked with a resistance gene. Facilitating the recombineering step with the CRISPR/Cas cleavage (applied either concomitantly or subsequently) aided the integration of long (>9 kbp) cassettes and was essential to insert unselectable genes (e.g., *gfp*) or to obtain double co-integrants of selectable gene cassettes in a single step. We displayed the reproducible and controllable copy number amplification of IS*3*-carried transgenes by expressing the respective transposase in trans. Our work also demonstrated the increased stability of chromosomal transgenes in the lack of selection, compared to their plasmid-borne counterparts. The flexibility of this strategy, called inPOSE, is showcased using the IS*3* element, but is most likely applicable to other bacterial ISes as well. For those who wish to use IS*3* for this purpose in strains harboring no IS*3* copies, we developed tools to rapidly introduce a marked copy of IS*3* into the chromosome by transposition.

## Figures and Tables

**Figure 1 microorganisms-10-00236-f001:**
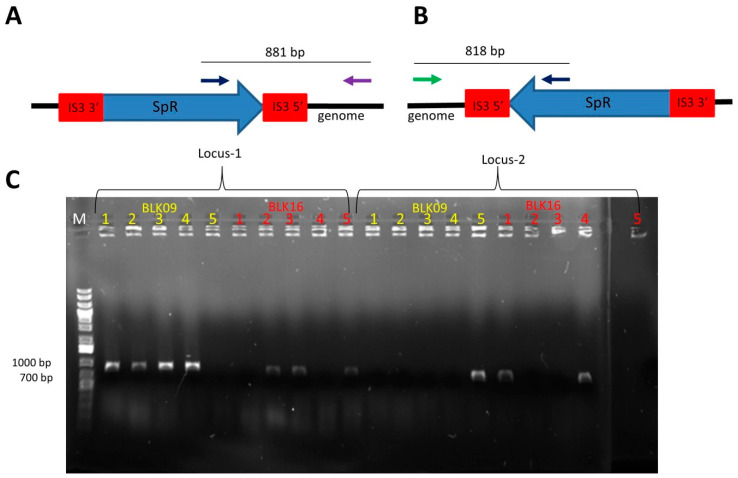
Colony PCR to verify chromosomal SpR integration into IS*3* upon recombineering. (**A**) Expected genetic arrangement and PCR product in the case of successful SpR integration at locus-1. (**B**) Expected genetic arrangement and PCR product in the case of successful SpR integration at locus-2. Black arrow: primer SmFw; purple arrow: primer IS3flanking1; green arrow: primer IS3flanking2; SpR: spectinomycin resistance gene. Red boxes depict IS*3*-homologies. (**C**) Gel electrophoresis of colony-PCR reactions made for screening of 5 *E. coli* BLK09 and 5 *E. coli* BLK16 colonies obtained upon recombineering. Primer pairs either specific for locus 1-integration or for locus 2-integration were used in both cases. M: GeneRuler 1 kb Plus DNA Ladder (Thermo Scientific).

**Figure 2 microorganisms-10-00236-f002:**
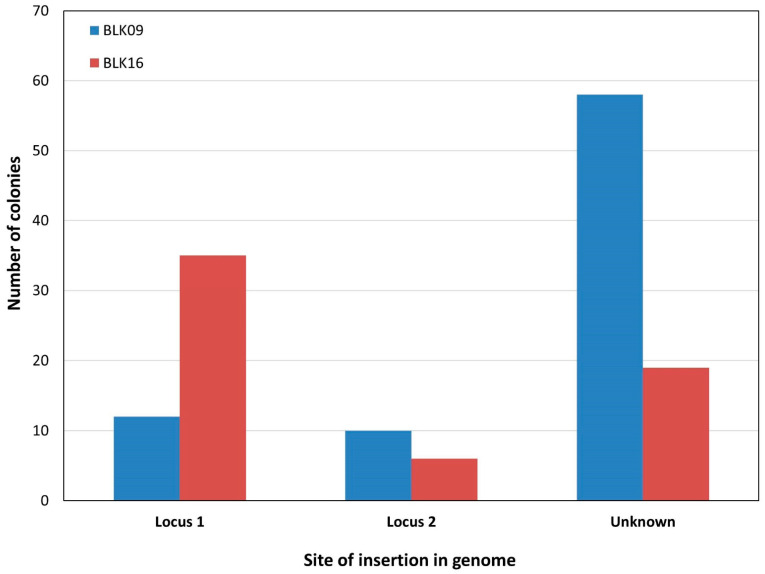
Combined result of recombineering experiments integrating the *vioABCDE*_Km or *vioABCDE*_SpR cassette into a chromosomal IS*3*. The number of colonies harboring a PCR-verified integration at locus 1, locus 2, or displaying no PCR-verified integration (“unknown”) are shown for *E. coli* BLK09 (blue) and *E. coli* BLK16 (red). The displayed results derive from a set of four similar experiments (two for each strain).

**Figure 3 microorganisms-10-00236-f003:**
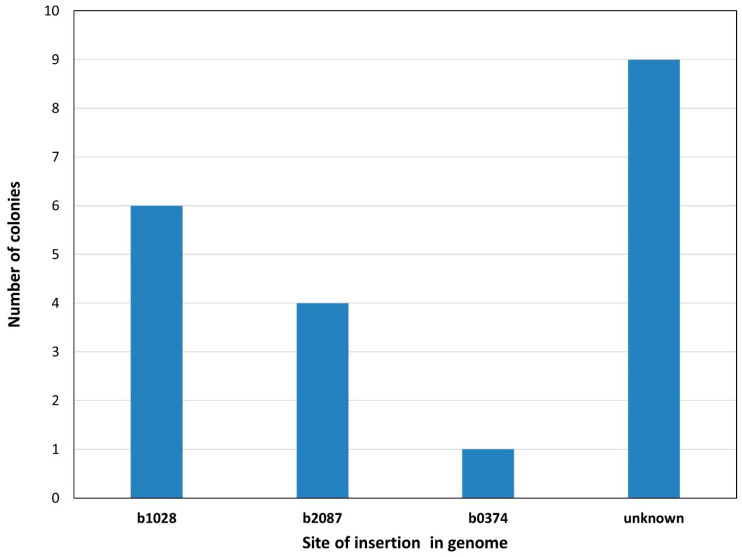
Result of recombineering experiments integrating the *vioABCDE*_KmR cassette into a chromosomal IS*3* of *E. coli* MDS16. The number of colonies harboring a PCR-verified integration at either of three loci (b1028, b2087 or b03741) or displaying no PCR-verified integration (“unknown”) are shown. An exemplary experimental outcome is displayed from a set of two similar experiments.

**Figure 4 microorganisms-10-00236-f004:**
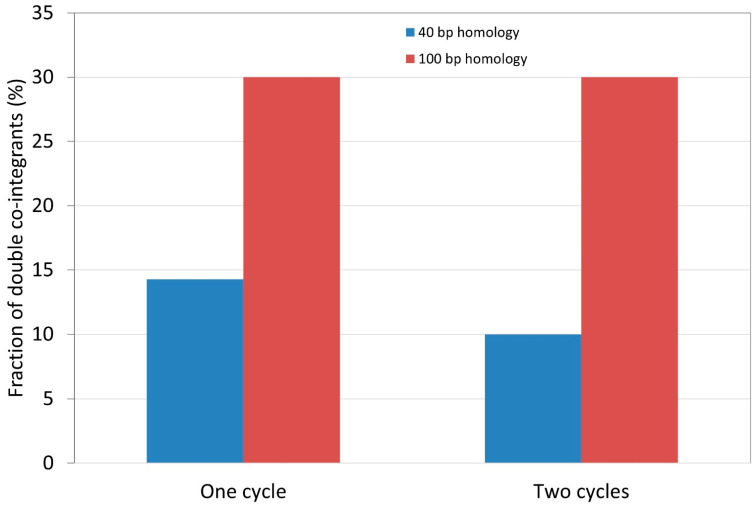
Efficiency of engineering double co-integrants. The IS*1* elements of *E. coli* MDS39R2 were targeted by recombineering followed by consecutive pCas9IS1-mediated counterselection. The fraction of colonies harboring an SpR cassette within both chromosomal IS*1* elements are shown after one or two cycles of recombineering and one round of counterselection. Blue bars depict the use of short (40 bp) and red bars depict the use of long (100 bp) homologies. An exemplary experimental outcome is displayed from a set of two similar experiments.

**Figure 5 microorganisms-10-00236-f005:**
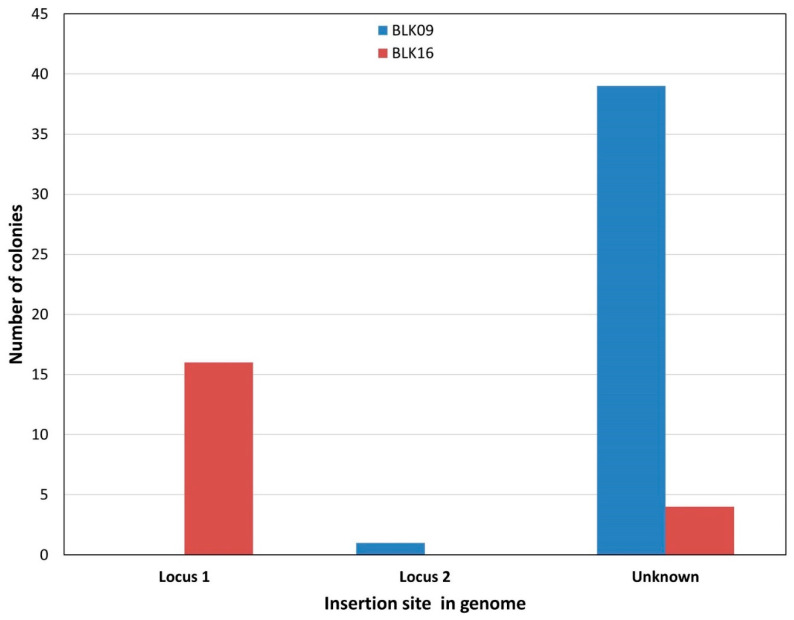
Integration of *vioABCDE*_KmR cassettes into chromosomal IS*3* elements using recombineering followed by pCas9IS3-mediated counterselection. The number of colonies harboring a PCR-verified integration at locus 1, locus 2, or displaying no PCR-verified integration (“unknown”) are shown for *E. coli* BLK09 (blue) and *E. coli* BLK16 (red). The yields of recombinants were 4 and 30 recombinants/ng DNA for BLK09 and BLK16, respectively. An exemplary experimental outcome is displayed from a set of four similar experiments (two for each strain).

**Figure 6 microorganisms-10-00236-f006:**
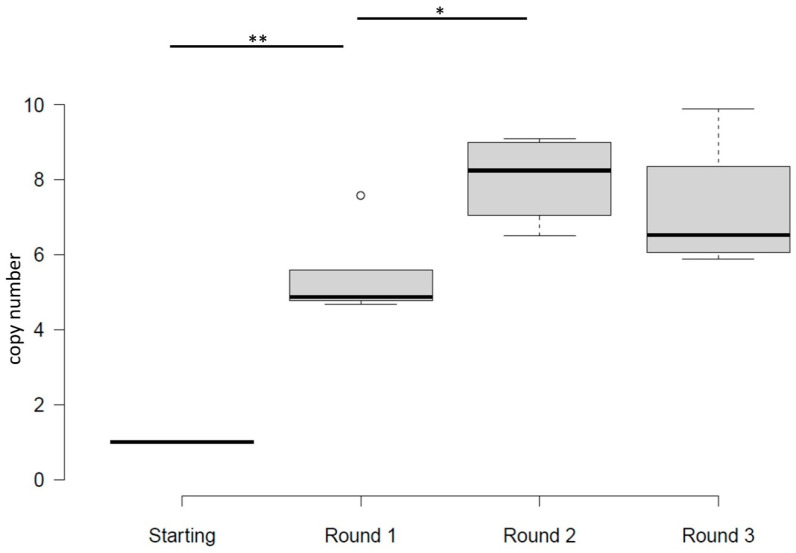
Copy numbers of the SpR gene of *E. coli* BLK16_IS*3*::*vioABCDE*_SpR detected by ddPCR after 1, 2 or 3 rounds of IS*3* transposase induction. Center lines show the medians; box limits indicate the 25th and 75th percentiles as determined by R software; whiskers extend 1.5 times the interquartile range from the 25th and 75th percentiles, and outliers are represented by dots. *n* = 1, 5, 4, 4 sample points. * *p* < 0.05 with two-tailed, unpaired *t*-test; ** *p* < 0.002 with two-tailed, one sample *t*-test.

**Figure 7 microorganisms-10-00236-f007:**
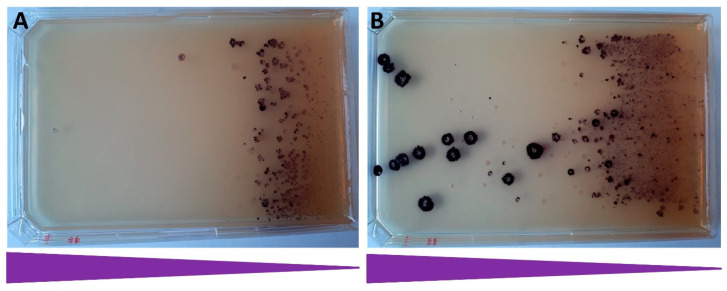
The effect of one round of IS*3* transposase induction on the Sp-resistance of BLK09IS*3*::*vioABCDE*_SpR (locus 2) colonies. Note the intense purple colors of the colonies caused by violacein. (**A**) Uninduced cells; and (**B**) cells treated with aTc to induce transposase expression. The triangles represent the gradient of Sp concentration within the medium.

**Figure 8 microorganisms-10-00236-f008:**
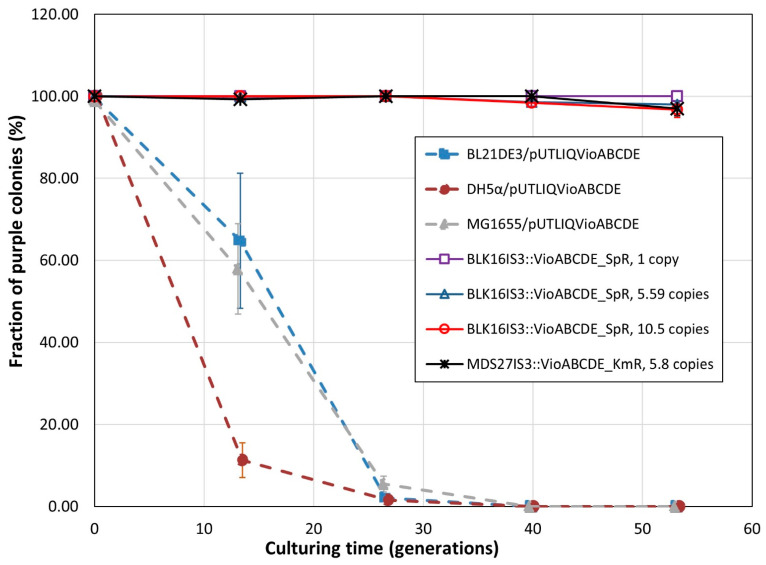
Fractions of violacein-producing cells within bacterial cultures grown in the lack of antibiotic selection. Dashed lines mark strains carrying plasmid pUTLIQvio_ABCDE, solid lines mark strains carrying an IS*3*::*vioABCDE* cassette on their chromosomes, as indicated in the legend. All values are means of three biological replicates. Error bars mark the standard error of the means.

**Figure 9 microorganisms-10-00236-f009:**
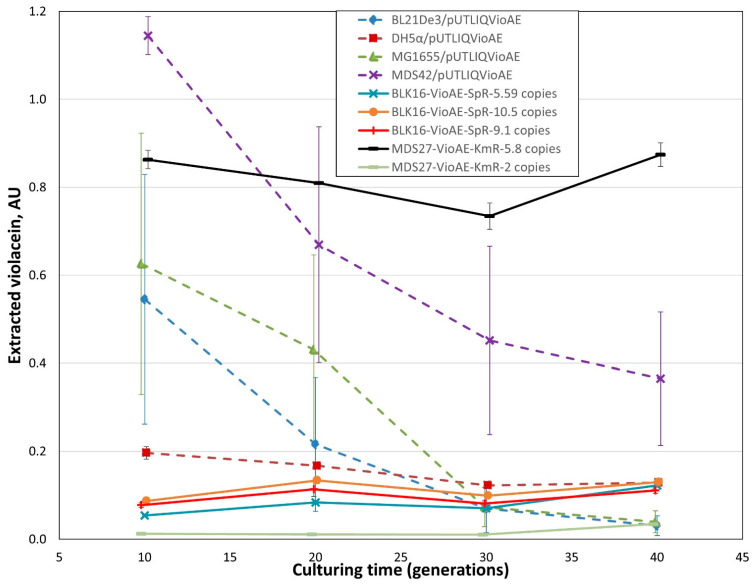
Violacein levels of liquid bacterial cultures grown in the lack of antibiotic selection. Dashed lines mark strains carrying plasmid pUTLIQ_*vioABCDE*, solid lines mark strains carrying an IS3::*vioABCDE* cassette on their chromosomes, as indicated in the legend. All values are means of three biological replicates. Error bars mark the standard error of the mean. The statistical analysis of the violacein content measured at generation 40 is shown in Appendix A.

**Figure 10 microorganisms-10-00236-f010:**
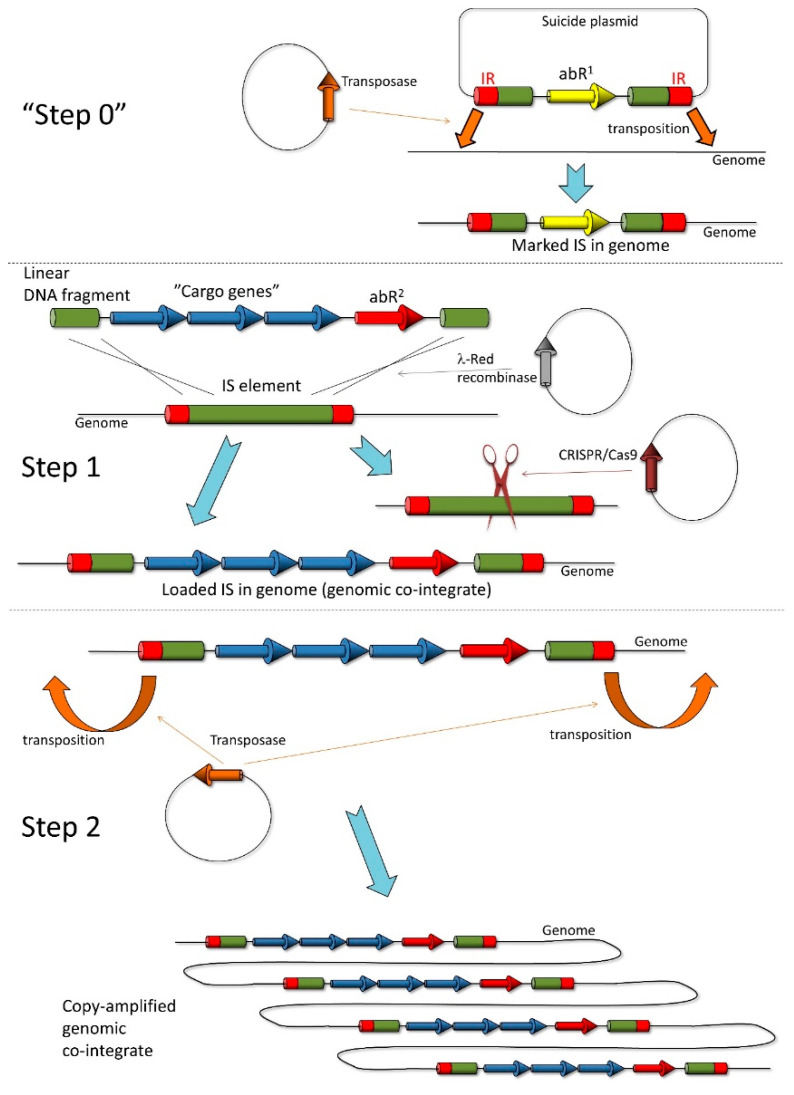
The inPOSE protocol. Step 1 is the entry of a gene or operon of interest into an IS element residing on the host genome by recombineering, facilitated by the λ-Red recombinase enzymes. CRISPR/Cas-mediated cleavage of the wt IS element(s) enforces selection for the recombinants and (if applied concomitantly) the facilitation of the recombination event. In Step 2, the genomic co-integrant (i.e., the loaded IS) is copy-amplified by the transposase corresponding to the IS, expressed in trans. “Step 0” is an optional accessory step that can transpose the marked IS element into the genome of host cells chosen for chromosomal transgene cloning. Green boxes: the targeted IS element (or segments thereof); red boxes: inverted repeats (IRs) of the IS; yellow and red arrows: two different antibiotic resistance genes (abR^1^ and abR^2^, respectively); blue arrows: the transgenes to be integrated; orange arrows: transposase gene of the targeted IS; gray arrow: λ-Red recombinase genes; brown arrow: CRISPR/Cas genes; open thin black lines: host bacterial genome; closed thin black lines: circular plasmids.

**Table 1 microorganisms-10-00236-t001:** *E. coli* strains modified in this study.

*E. coli* Strain	Reference	GenBank Acc. No.
MG1655	[36]	NC_000913.3
DH5a	[37]	NZ_CP080399
MDS16	this work	-
MDS27	this work	-
MDS30	this work	-
MDS39R2	this work	-
MDS42	[38]	AP012306
BL21(DE3)pLysE	[39]	NC_012947.1
BLK09	[40]	CP014641
BLK16	[40]	CP014642

**Table 2 microorganisms-10-00236-t002:** Plasmids constructed for this study.

Plasmid	Origin	Resistance	Function
pKDsg-IS1	pSC101*	Sp	expression of λ-Red recombinases and a crRNA targeting IS*1*
pKDsg-IS3	pSC101*	Sp	expression of λ-Red recombinases and a crRNA targeting IS*3*
pST76AIS1X	pSC101*	Ap	carrying two homology boxes of IS*1*, without inverted repeats
pST76AIS3X	pSC101*	Ap	carrying two homology boxes of IS*3*, without inverted repeats
pST76AIS1X::*vioABCDE*	pSC101*	Ap	carrying two homology boxes of IS*1* (without inverted repeats), flanking the *vioABCDE* operon
pST76AIS3X::*vioABCDE*	pSC101*	Ap	carrying two homology boxes of IS*3* (without inverted repeats), flanking the *vioABCDE* operon
pST76AIS1X::*vioABCDE*_SpR	pSC101*	Ap, Sp	carrying two homology boxes of IS*1* (without inverted repeats), flanking the *vioABCDE* operon and the SpR gene
pST76AIS3X::*vioABCDE*_SpR	pSC101*	Ap, Sp	carrying two homology boxes of IS*3* (without inverted repeats), flanking the *vioABCDE* operon and the SpR gene
pST76AIS1X::*vioABCDE*_KmR	pSC101*	Ap, Km	carrying two homology boxes of IS*1* (without inverted repeats), flanking the *vioABCDE* operon and the KmR gene
pST76AIS3X::*vioABCDE*_KmR	pSC101*	Ap, Km	carrying two homology boxes of IS*3* (without inverted repeats), flanking the *vioABCDE* operon and the KmR gene
pZA31insAB’tetR	p15A	Cm	expression of IS*1* transposase *insAB*’, encoded within a single frame
pSTinsAB’tetR	pSC101*	Cm	expression of IS*1* transposase *insAB*’, encoded within a single frame (heat-sensitive rep origin)
pZA31tnp3tetR	p15A	Cm	expression of codon optimized IS*3* transposase *insEF*’, encoded within a single frame
pSTtnp3tetR	pSC101*	Cm	expression of codon optimized IS*3* transposase *insEF*’, encoded within a single frame (heat-sensitive rep origin)
pSG78A_full_IS3::SpR	R6K	Ap, Sp	carrying two homology boxes of IS*3* (including inverted repeats), flanking the SpR marker
pCas9IS1	p15A	Cm	expression of the full CRISPR/Cas9 machinery targeting IS*1*
pCas9IS3	p15A	Cm	expression of the full CRISPR/Cas9 machinery targeting IS*3*

pSC101*: stands for the thermosensitive version of the pSC101 origin of replication. Cm: chloramphenicol; Ap: ampicillin; Km: kanamycin; Sp: spectinomycin.

**Table 3 microorganisms-10-00236-t003:** The fraction of colonies obtained on the selective plate that carried the co-integrant at either IS*3* locus, as verified by PCR.

	Selection Marker Used	
	SpR	KmR
BLK09	20%	65%
BLK16	75%	50%
MDS27	25%	45%
MDS16	ND	55%

**Table 4 microorganisms-10-00236-t004:** Properties of *E. coli* BLK16IS*3*::*vioABCDE*_SpR derivatives after the indicated rounds of IS3 transposase induction.

Strain Code	Strain Name	SpR Copy Number by ddPCR	Violacein Content (AU)	Relative Violacein Content ^1^	Relative Sequence Coverage of *vioABCDE* by WGS ^2^
B0	BLK16IS3::*vioABCDE*_SpR, starting	0.98	0.06	1	1
B1	BLK16IS3::*vioABCDE*_SpR, 1st round	5.59	0.2926	4.88	5.71
B2	BLK16IS3::*vioABCDE*_SpR, 2nd round	9.10	0.7847	13.08	11.32
B3	BLK16IS3::*vioABCDE*_SpR, 3rd round	26.11	0.8113	13.52	12.51

SpR: spectinomycin resistance; ddPCR: droplet digital polymerase chain reaction; AU: arbitrary unit, based on OD_585_ measurement; WGS: whole genome sequencing; ^1^ Relative to strain B0. ^2^ Ratio of sequence reads obtained +/− 10 nt of peak border.

## Data Availability

All genome sequencing data has been submitted to the Sequence Read Archive (SRA) of NCBI, BioProject ID: PRJNA798701.

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
