# Peer review of "inPOSE: A Flexible Toolbox for Chromosomal Cloning and Amplification of Bacterial Transgenes"

_microorganisms, 2022, doi:10.3390/microorganisms10020236_

Round 1

Reviewer 1 Report

Review:

inPOSE: a flexible toolbox for chromosomal cloning and amplification of bacterial transgenes

Ranti Dev Shukla, Ágnes Zvara, Ákos Avramucz, Alona Yu. Biketova, Akos Nyerges, László G. Puskás, Tamás Fehér

Synopsis:

The authors develop an innovative way to integrate transgenes at single copy into IS elements located in the E coli chromosome followed by amplifying the transgene to multiple copies using an inducible copy/paste transposase.

Comments and notes for improvement:

  1. Perhaps a brief note can be incorporated why the inverted repeats were not included during the generation of the recombineering templates, for clarification towards the novice?
  2. Figure 2: can a small red and blue box, indicating BLK09 amd BLK16, respectively, be added to the figure, for clarification?
  3. Figure 4: can a small red and blue box, indicating 40 bp homology amd 100 bp homology, respectively, be added to the figure, for clarification?
  4. Figure 5: can a small red and blue box, indicating BLK09 amd BLK16, respectively, be added to the figure, for clarification?
  5. Figure 7. legend: please indicate that aTc induction results in IS3 transposase activation, for clarification.
  6. Figure 8, legend: please indicate that pUTLIQvio_ABCDEis a plamid, for clarification.
  7. Figure 9, legend: please indicate that pUTLIQvio_ABCDEis a plamid, for clarification.
  8. Can some examples be given where this method would be very useful?

Conclusion:

The method presented by the authors is an innovative way to increase the copy number of a transgene integrated in the chromosome. I am happy to accept this work for publication after the incorporation of the above comments

Author Response

Response to Reviewer 1

We cordially thank the Reviewer for the positive opinion on our manuscript, as well as the helpful comments. Our responses are as follows:

“Perhaps a brief note can be incorporated why the inverted repeats were not included during the generation of the recombineering templates, for clarification towards the novice?”

Response: We added the following sentence to section 3.1, line 362: “Note that the linear fragments did not harbor the inverted repeats of the IS to avoid binding of the transposase enzyme and potentially causing fragment entry into the genome by random transposition.”

“Figure 2: can a small red and blue box, indicating BLK09 amd BLK16, respectively, be added to the figure, for clarification?

Figure 4: can a small red and blue box, indicating 40 bp homology amd 100 bp homology, respectively, be added to the figure, for clarification?

Figure 5: can a small red and blue box, indicating BLK09 amd BLK16, respectively, be added to the figure, for clarification?”

Response: We added the missing boxes to aid the interpretation of the charts.

“Figure 7. legend: please indicate that aTc induction results in IS3 transposase activation, for clarification.”

Response: We modified the relevant sentence in the figure legend to the following: “(A) Uninduced cells, (B) cells treated with aTc to induce transposase expression.”

“Figure 8, legend: please indicate that pUTLIQvio_ABCDE is a plasmid, for clarification.

Figure 9, legend: please indicate that pUTLIQvio_ABCDE is a plasmid, for clarification.”

Response: We modified both figure legends to contain “plasmid pUTLIQvio_ABCDE”.

“Can some examples be given where this method would be very useful?”

Response: We added the following paragraph before the ultimate paragraph of the Discussion:

“The potential utility of inPOSE is easy to envision: numerous examples are available in the literature requiring the expression of RNA- or protein-encoding genes from the bacterial chromosome. For example, E. coli strains carrying an elevated number of rRNA operons [56] or extra copies of rare tRNA genes [57] have been engineered for fundamental and applied scientific purposes, respectively. A more common need is to express heterologous enzymes or entire enzymatic pathways from genes inserted into the bacterial chromosome [22,23,58,59]. A number of such strains published to date displayed improved genetic stability even in the lack of chemical selection, a crucial feature for the fermentation industry [22,23]. The improvement of production titers by copy-number amplification has also been demonstrated [22,23]. In certain cases, proper balancing of the heterologous pathway was granted by the isolated alteration of certain enzyme-encoding gene’s copy number [24,58,59]. The inPOSE protocol fits into the line of such experiments, providing a flexible tool for the integration and amplification of heterologous pathway-encoding gene constructs. Ultimately, the use of multiple IS elements within one engineered strain could allow combinatorial alteration of the copy numbers of genes constituting the pathway, al-lowing the selection for the best producer using a high-throughput assay.”

Reviewer 2 Report

In the fuigures #2, 3, 4, 5 results of single experiments are shown. Were they really single or the best (or the one of) result of the series represented? It is not clear, if the results were reproducible?

In the figures #8 and 9 mean values are shown? Why don’t you represent the dispersion of the data in the form of SD or smth like that?

Author Response

Reviewer 2

We thank the reviewer for spending time and effort with our manuscript, and highly appreciate the positive marks and the critical comments.

“In the figures #2, 3, 4, 5 results of single experiments are shown. Were they really single or the best (or the one of) result of the series represented? It is not clear, if the results were reproducible?”

Response: The indicated figures display typical results from a set of similar (but not necessarily alike) parallel experiments. Since the trends shown in these charts are not used to draw any specific conclusions, the parallel datasets were not subjects of statistical analysis. We added the following sentence to the indicated figures’ legends for clarification: “An exemplary experimental outcome is displayed from a set of ‘N’ similar experiments.“

“In the figures #8 and 9 mean values are shown? Why don’t you represent the dispersion of the data in the form of SD or smth like that?”

Response: We added error bars to the charts in Figure 8 and 9, representing the standard error of the means. We noted this information in the figure legends, too.

Reviewer 3 Report

The manuscript entitled inPOS a flexible toolbox for chromosomal cloning and amplification of bacterial transgenes describes a protocol which aims to introduce multiple copies of a gene in the bacterial chromosome.

The manuscript is well written and the necessity of such a protocol is argumented in the introduction and conclusion. The experimental design is well described and the final protocol succeeds to introduce multiple copies of either an antibiotic resistance gene or a long cassette. However the expression of violacein from this long cassette is not well conserved along the experiments.

Comments

I have one concern about the IS1-mediated integration that displays a sub-optimal performance in this work: the authors developed their protocol by using IS3 but I wonder whether the inserted sequence may be the cause of the suboptimal performance. Do the authors have any arguments to answer to this question? Or can they show experimentally that this is not the case?

In addition, in figure S5 where the gfp gene is inserted in IS3 or IS1 and the fluorescence of the strains measured, if the p value shows a significant difference of expression without or with the gfp gene in both strains, for the IS3 insertion the increase in fluorescence is only of 30%. In contrast the increase of expression in IS1 insertion is 200%. Is the expression of the gene depending of the site of insertion? As the purpose of these constructs is the expression of genes, it is of primary importance to know whether it does or not depend on the insertion site.

Author Response

Reviewer 3

“The manuscript entitled inPOS a flexible toolbox for chromosomal cloning and amplification of bacterial transgenes describes a protocol which aims to introduce multiple copies of a gene in the bacterial chromosome.

The manuscript is well written and the necessity of such a protocol is argumented in the introduction and conclusion. The experimental design is well described and the final protocol succeeds to introduce multiple copies of either an antibiotic resistance gene or a long cassette. However the expression of violacein from this long cassette is not well conserved along the experiments.”

Response: We express our gratitude to the Reviewer for the positive remarks and for raising interesting points of discussion.

Comments

“I have one concern about the IS1-mediated integration that displays a sub-optimal performance in this work: the authors developed their protocol by using IS3 but I wonder whether the inserted sequence may be the cause of the suboptimal performance. Do the authors have any arguments to answer to this question? Or can they show experimentally that this is not the case?”

Response: The targeting of IS1 was problematic from several aspects, most importantly, the violacein operon was non-functional upon integration, and the amplification of an integrated resistance marker was poorly reproducible. The two malfunctions may or may not be related. We note that the resistance gene and gfp both seemed to be expressed properly from integrated IS1 elements. Indeed, one possibility could be an unexpected interaction between the cargo sequence (specifically, the vio operon) and the remainder of the IS1. This could be, for example, a transcriptional inhibition of the operon caused by secondary DNA structures formed by the IRs and the operon. Alternatively, the lambda-Red recombinases may also have caused unexpected rearrangements of the operon in the presence of IS1 IRs. We regret, but we do not possess any experimental data to argue in favor of either mechanism, or other possible mechanisms. Clarification of this issue could be an interesting research topic, it is however beyond the scope of our current project.

“In addition, in figure S5 where the gfp gene is inserted in IS3 or IS1 and the fluorescence of the strains measured, if the p value shows a significant difference of expression without or with the gfp gene in both strains, for the IS3 insertion the increase in fluorescence is only of 30%. In contrast the increase of expression in IS1 insertion is 200%. Is the expression of the gene depending of the site of insertion? As the purpose of these constructs is the expression of genes, it is of primary importance to know whether it does or not depend on the insertion site.”

Response: We indeed assume that the genetic environment on the chromosome is at least partly responsible for the observed difference in the level of GFP expression. Earlier works have reported the strong dependence of transgene expression on the site of integration. For example, Yin et al., [Appl Microbiol Biotechnol (2015) 99:5523] found nearly two fold differences in transgene expression measured at various loci. However, we cannot exclude the additional effect of the genetic MICROenvironment, i.e. the IS elements themselves on the levels of the gfp transcription. A further experiment placing the IS1::gfp and the IS3:: gfp into exactly the same locus, and comparing expression levels could resolve this issue. Currently however, such experiments are beyond our time allowance provided for this revision.

Round 2

Reviewer 3 Report

The manuscript is correct despit the fact that the authors did'nt answer to the previous remarks as they believed it is beyond the scope of their current project. Anyway it is a complete protocol and it's potential utility is well describe now in the conclusions.